# Semantic-aware Representation Learning for Homography Estimation

## ABSTRACT

Homography estimation is the task of determining the transformation from an image pair. Our approach focuses on employing detector-free feature matching methods to address this issue. Previous work has underscored the importance of incorporating semantic information, however there still lacks an efficient way to utilize semantic information. Previous methods suffer from treating the semantics as a pre-processing, causing the utilization of semantics overly coarse-grained and lack adaptability when dealing with different tasks. In our work, we seek another way to use the semantic information, that is semantic-aware feature representation learning framework. Based on this, we propose SRMatcher, a new detector-free feature matching method, which encourages the network to learn integrated semantic feature representation. Specifically, to capture precise and rich semantics, we leverage the capabilities of recently popularized vision foundation models (VFMs) trained on extensive datasets. Then, a cross-images Semantic-aware Fusion Block (SFB) is proposed to integrate its fine-grained semantic features into the feature representation space. In this way, by reducing errors stemming from semantic inconsistencies in matching pairs, our proposed SRMatcher is able to deliver more accurate and realistic outcomes. Extensive experiments show that SRMatcher surpasses solid baselines and attains SOTA results on multiple real-world datasets. Compared to the previous SOTA approach GeoFormer, SRMatcher increases the area under the cumulative curve (AUC) by about 11% on HPatches. Additionally, the SRMatcher could serve as a plug-and-play framework for other matching methods like LoFTR, yielding substantial precision improvement.

## CCS CONCEPTS

• **Computing methodologies** → **Computer vision**.

## KEYWORDS

homography estimation, semantic, feature matching

## 1 INTRODUCTION

Homography estimation, often referred to as perspective transformation or planar projection from source image to target image as Figure 1 shown, is a core issue in the fields of computer vision and robotics [13, 41]. This process entails calculating a matrix to align points between two images captured from distinct viewpoints. This

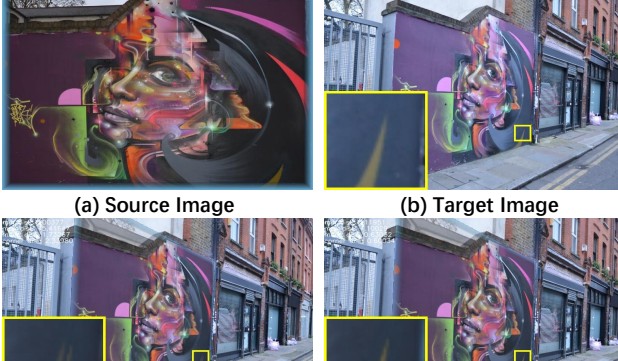

(a) Source Image     (b) Target Image

(c) MESA warp merged(error=2.33)     (d) Ours (error=0.60)

**Figure 1: Homography transformation results by our proposed SRMatcher and MESA [42]. The blue box line was added artificially to highlight the range of homologous transformations. The yellow boxes show the real scenes cropped in the target image. (c) and (d) are generated by superimposing the warped source image on the target image, showing our SRMatcher acquiring more accurate and realistic outcomes.**

estimation is pivotal for numerous applications, including including image/video stitching [11, 40], camera calibration [44], HDR imaging [9], and SLAM [23, 45]. Feature matching techniques are widely used to establish feature correspondences in tasks related to homography estimation. The traditional matching methods [22, 29] rely on the quality of keypoints, and can sometimes falter in challenging scenarios such as low texture and blurred [31]. A novel detector-free matching methods have surfaced as more dependable alternatives. These techniques generate dense feature maps without the need for feature detection and execute dense matching on a pixel-wise basis [3, 4, 15, 20, 32]. Existing methods typically extract local features related to position and context but overlook the semantic information of matching pairs, which is crucial for feature matching.

Previous work [42, 43] such as MESA do consider semantics, these methods view the matching issue as a search problem, searching corresponding matching points for specific source points in the target image. The introduction of semantics aims to narrow the search space, by establishing area matches the initial search space for point matching is confined to the matched area. This method resembles the human cognitive process, humans first identify matched areas using semantic information and subsequently search for corresponding points within those areas. However, their work faces two notable problems: **1) The first problem** is these methods restrict the application of semantic priors to coarse-grained scenarios, failing to fully leverage the knowledge that semantic extraction networks can offer. The goal of incorporating semantic priors into the

matching network is to reduce erroneous matches by constraining the semantic consistency of the matching results. However these area matching method merely narrows down the matching area, and semantics can not guide the matching result in subsequent steps, failing to achieve the initial goal. **2) For the second problem**, the interaction between semantic information and matching is designed for particular approaches. When dealing with different tasks or data types, processing methods such as area size and quantity need optimization for particular applications. This means these patch-wise method suffer from a lack of adaptability and generalize issues, limiting the ability of the algorithm to understand the overall scene.

We identify the key driver of these two problems as the ill-use of the semantic information, specifically, treating semantics as merely a pre-processing step. These methods initially identify semantic area matches based on semantic segmentation and then acquire precise correspondences within these regions using a standard point matcher. This process is intuitive and closely resembles human thought processes [33, 43]. However, for the human cognitive system, an implicit prerequisite in the task of feature matching is that the features being matched should share the same semantics [10, 35, 36]. Based on this, we propose another way to use the semantic information that encourages the network to learn semantic-aware representation.

In this paper, we propose SRMatcher, the first semantic-aware representation learning framework which consists of two essential parts: **(a)** Semantic Extractor. **(b)** Semantic-aware Fusion Block(SFB). The key challenge to achieving the semantic-aware representation is *how to generate similar representations for semantically similar points in image pairs*. **For (a)**, we propose to use the DINOv2 [24] as the semantic extractor to acquire richer and more diverse semantic information. This considers the outstanding performance of vision foundation models (VFMs) in tasks related to semantics. **For (b)**, we develop a Semantic-aware Fusion Block (SFB) to utilize fine-grained semantic features to contribute to matching quality. Different from previous works [17, 37, 38], SFB executes cross-images feature fusion which means image features should not only integrate their own semantic information but also consider the semantics across images. This difference comes from considering the nature of the task, that is the image feature should be enhanced with semantic information of other images to find semantically identical points. In more detail, we employ a semantic-guide interactions block (SGIB) to establish the interactions between image features and semantic features within the SFB.

In summary, this work makes the following contributions:

- We explore a new way to integrate semantics into the feature matching model for homography estimation. For the first time, we propose SRMatcher, a novel detector-free feature matching method by integrating semantics from vision foundation models (VFMs) into the network.

- To enhance the semantic guidance for the matching model, we introduce the Semantic-aware Fusion Block (SFB) to conduct cross-images semantic feature fusion, whereby features are prompted to concurrently consider the semantics from multiple images.

- Extensive experiments demonstrate that SRMatcher achieves the SOTA performance on homography estimation tasks and other downstream vision tasks. Additionally, our SRMatcher can be seamlessly incorporated into various benchmarks for detector-free feature matching methods in a plug-and-play fashion.

## 2 RELATED WORK

### 2.1 Local Feature Matching

Local feature matching can be categorized into detector-based and detector-free methods. Detector-based methods operate in four stages: detecting keypoints, extracting local features for each keypoint, matching features based on content, and finally fitting a homography using the identified matches. SIFT [22] features exhibit invariance to rotation and scale. With the advent of deep learning, many learning-based methods have been proposed, to further enhance the robustness of descriptors under varying illumination and viewpoint changes, such as Superpoint [5], D2-Net [28]. SuperGlue [30] utilizes Transformer [34] to establish correspondences between two sets of local features generated by SuperPoint. The drawback of detector-based methods lies in their excessive reliance on the quality of keypoint detection, which is particularly unsatisfactory in low-texture and repetitive texture regions. Detector-free methods establish dense matching relationships directly between pixels, eliminating the need for keypoint detection. LoFTR [31] utilizes self- and cross-attention mechanisms on feature maps derived from CNN, producing matches progressively from coarse to fine detail. GeoFormer [20] using the RANSAC algorithm for attention region search, the computation scope of attention is confined to a specific region, enabling the use of a standard transformer for processing. While these approaches are methodologically robust, they fall short in capturing high-level contexts such as semantic information. They fail to explicitly represent the semantic correspondences within image pairs and lack clarity in their interpretability. Conversely, our method, through a Semantic-aware Fusion Block(SFB), can utilize fine-grained semantic-aware features, making our matching outcomes significantly more interpretable and accurate.

### 2.2 Semantics Feature Matching

Lately, methods guided by semantics have demonstrated the dependability of semantic priors. SGAM [43] incorporating semantics segmentation maps as an additional input, region matching based on identical semantics is conducted within the image before point matching. However, SGAM utilizing the direct semantic label results in the area matching's performance being highly sensitive to the precision of semantic labeling. Conversely, MESA [42] employs pure image segmentation for area matching, offering a more pragmatic approach and overcoming the limitations associated with direct semantics labels. Nevertheless, the semantic representation of this approach is coarse-grained, and semantic inconsistency still exists in the matched regions, when it comes to detailed point matching. In comparison, we explore the adaptive latest network frameworks that use a novel interplay between semantic priors and the original tasks, allowing the model to utilize rich semantic priors from a pre-trained encoder. Our approach integrates image appearance features with high-dimensional semantics, integrate

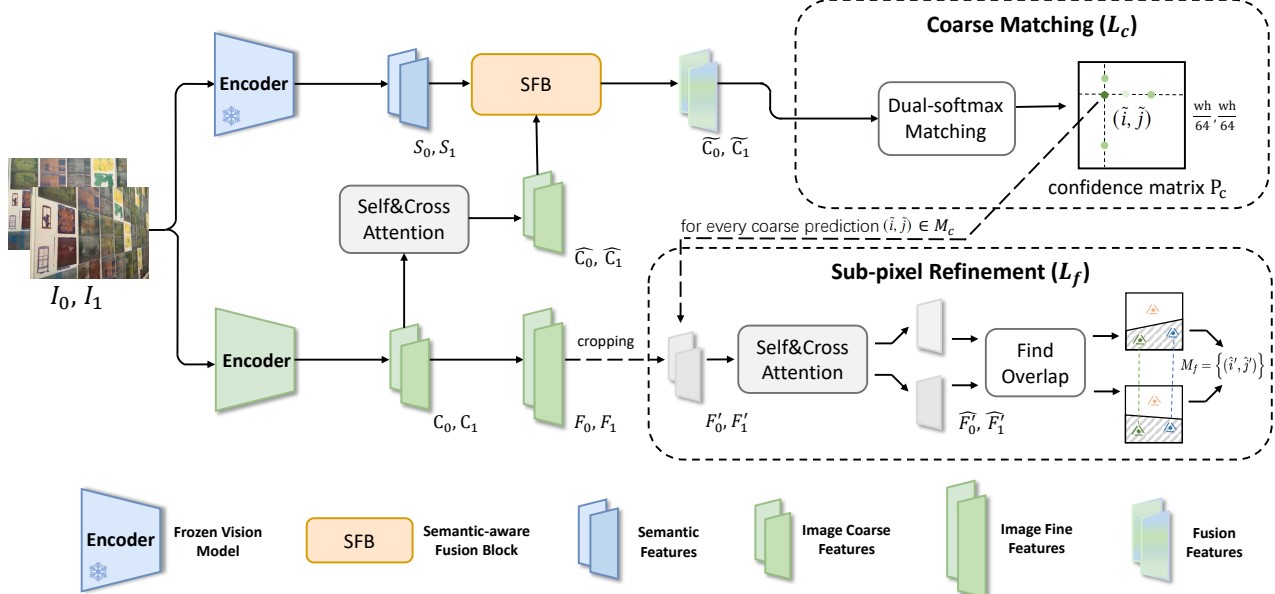

**Figure 2: Overview of our SRMatcher for detector-free local feature matching and followed by homography estimation. With a pretrained Semantic extract network, our SRMatcher utilizes fine-grained features to improve the matching results. The SFB enable interactions between image features $\hat{C}_0$ and $\hat{C}_1$ and semantic features $S_0$, $S_1$, produce the fusion features. The coarse matching block generates pixel-to-pixel matches $M_c$ at 1/8 scale. Subsequently, the $M_c$ input into the overlap-based fine matching to yield fine matches $M_f$ at 1/2 scale.**

fine-grained semantic into feature representation space. This allows our method to be a fully handcrafted, plug-and-play framework.

### 2.3 Vision Foundation Models

Leveraging extensive pre-training, foundational models in vision have garnered significant achievements in the field of computer vision, shows excellent performance in various downstream tasks [8, 17, 21, 42]. Inspired by the masked language modeling [6] from natural language processing, Masked Autoencoder (MAE) [12] utilizes an asymmetric encoder-decoder architecture for masked image modeling, facilitating the effective and efficient training of scalable vision transformer models, with MAE displaying outstanding fine-tuning capabilities in a variety of downstream tasks. CLIP [27] acquires image representations from the ground up using 400 million image-text pairs, showcasing remarkable zero-shot image classification skills. Through discriminative self-supervised learning at both the image and patch levels, DINOv2 [24] acquires versatile visual features applicable to a broad spectrum of downstream tasks, also exhibiting notable patch-matching proficiency that identifies semantic components executing similar functions across diverse objects or species. Motivated by these developments, our objective is to utilize more fine-grained semantics from DINOv2 to direct our SRMatcher.

### 3 APPROACH

#### 3.1 Overall Framework

As our approach is built upon LoFTR, we initially provide a concise description of this method. Given a pair of input images $I_0$ and $I_1$ with dimensions $w \times h$ in gray-scale, a 2D-CNN (ResNetFPN [19]) extracts two feature maps per image: a coarse-level feature map at 1/8 of the original size and a fine-level feature map at 1/2 of the original size. The coarse features $C_0$ and $C_1$ are input into a attention mechanisms to produce $\hat{C}_0$ and $\hat{C}_1$. Then the coarse matching module generates a pixel-to-pixel confidence matrix $P_c$. Coarse-level matches $M_c = \{(\tilde{i}, \tilde{j})\}$ are established by applying a threshold to $P_c$ and conducting a mutual nearest neighbor (MNN) search. Given $M_c$ and the fine-level features $F_0$ and $F_1$, the fine-matching module computes sub-pixel matches in the following way. For each pair $(\tilde{i}, \tilde{j}) \in M_c$, a square region of dimensions $w \times w$ centered around $\hat{i} = \tilde{i} \times 4$ is extracted from $F_0$, and a similar region centered around $\hat{j} = \tilde{j} \times 4$ is extracted from $F_1$. The optimal correspondence for pixel $\hat{i}$ within the $F_1$ region determines the set of sub-pixel matches $M_f$.

Inherited from LoFTR, we propose SRMatcher the semantic-aware feature representation learning framework. The comprehensive structure of our SRMatcher is shown in Figure 2. Starting with a pair of images, we initially process them through a CNN backbone to extract coarse and fine image features. And a frozen pre-trained semantic extractor from a large vision model to extract the semantics. Subsequently, the enhanced image features by self- and cross-attention undergo processing in our SFB module to integrate with fine-grained semantic features. Following this, coarse matching is utilized to establish patch-level matches. Ultimately, the overlap based fine-matching is employed to predict fine-level matches. With the semantically informed features, the network can perform semantic-aware feature matching with our

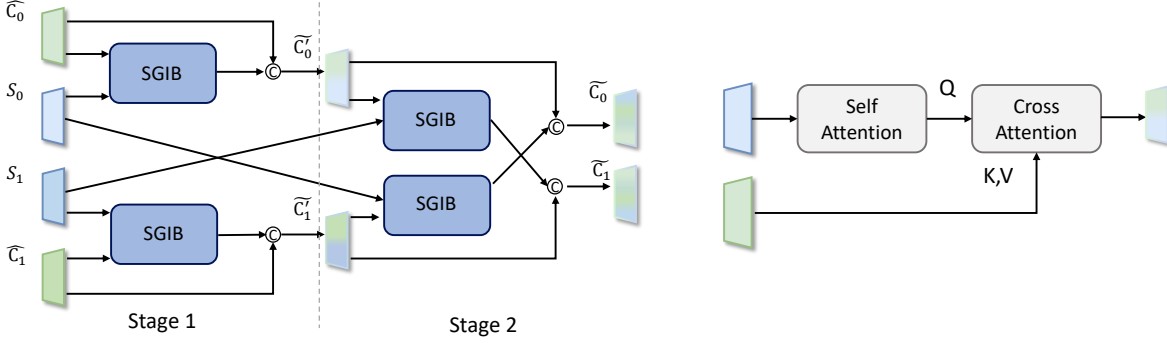

(a) Semantic-aware Fusion Block(SFB)  (b) Semantic-guide Interactions Block(SGIB)

Figure 3: Architecture of the (a)semantic-aware fusion block(SFB) and (b)semantic-guide interactions block(SGIB). The SFB fuses image features and semantic features across images. Inside SFB, SGIB computes cross-attention that the image features as key K / value V and the semantic feature as query Q.

proposed framework. The problem definition of the semantic-aware representation learning framework is as follows:

Given a pair of image $I_0$ and $I_1 \in \mathbb{R}^{H \times W \times 3}$ with width $W$ and height $H$. Combining with fine-grained semantic features, the matching process can be modeled as follow, first:

$$S_i = F_{semantic}(I_i; \theta_s), i = 0, 1 \qquad (1)$$

where $S_i$ represents the fine-grained semantic features. $F_{semantic}$ denotes the pre-trained semantic extraction network, with $\theta_s$ being frozen during the training stage. $S_i$ is then utilized as input:

$$M_h = F_{match}(I_i, S_i; \theta_m) \qquad (2)$$

where $M_h$ is the matched result and $F_{match}$ represents the matching network. Throughout the training stage, $\theta_m$ is updated by minimizing the objective function with the collaboration of $S_i$, while $\theta_s$ is frozen:

$$\hat{\theta_m} = argminL(\hat{M_h}, M_h, S_i) \qquad (3)$$

where $\hat{M_h}$ is the ground truth, $L(\hat{M_h}, M_h, S_i)$ is the objective function of semantic-aware matching model. The details of our semantic-aware feature representation learning framework will be elaborated in the following section.

## 3.2 Semantic-aware Fusion Block

When enhancing image features using semantic features, it's crucial to address the discrepancies between the two sources. To mitigate this, we introduce the SFB module to refine the image feature maps, as depicted in Figure 3 (a). Serving as a bridges between the semantic net and the matching net, the SFB modules create connections between these two distinct tasks.

The efficacy of semantic extraction is crucial for the semantic-aware feature representation learning framework. The motivation is that fine-grained semantics will enable the network to generate more precise matching results. The advanced vision foundation models (VFMs), renowned for their robust representational prowess, are ideal for this task. Particularly, the DINOv2 model stands out for its effectiveness and versatility across diverse applications [21, 39], making it a preferred choice for semantic extraction. For our

purposes, we select the pretrained DINOv2 model as our semantic extractor to obtain pixel-level semantic information.

Upon acquiring the semantics $S_i$, we utilize the proposed Semantic-aware Fusion Block (SFB) to integrate semantic features into the matching model. With the enhanced image features of coarse-level $\hat{C_0}$ and $\hat{C_1}$, our goal is to warp the semantic features from the images to the matching model. Within the SFB, we have designed a cross-images feature fusion strategy, as image features are to be augmented not only based on their own semantic information but also taking into account the semantic information from across different images. For the input features $S_i$ and $\hat{C_i}$, we propose a semantic-guide interactions block (SGIB) within SFB to more effectively integrate fine-grained semantics features with image features.

**Semantic-guide Interactions Block (SGIB).** In the first stage of SFB, the image feature and semantic features from the same image are fused. The SGIB module conducts a pixel-level interaction between image features $\hat{C_i}$ and semantic features $S_i$ and output the refined feature map $\tilde{C_i}'$. As shown in Figure 3 (b), given the semantic features $S_i$, first go through a self-attention(SA) module and then supplied to the cross-attention module as query $Q_i$, then the image features $\hat{C_i}$ are identified as the keys $K_i$ and values $V_i$:

$$Q_i = \mathcal{SA}(S_i)$$
$$\tilde{C_i}' = Concat(\hat{C_i}, (Softmax(Q_i K_i^T / \sqrt{C})) V_i) \qquad (4)$$

where C denote the channel and the Concat represent the concatenate operation. And then at the second stage of SFB, the image feature and semantic features from the different image are fused. The calculation process is similar to the first stage, the keys $K_i'$ and values $V_i'$ are produced by $\tilde{C_i}'$:

$$Q_j' = \mathcal{SA}(S_j), j = 0, 1$$
$$\tilde{C_i} = Concat(\tilde{C_i}', (Softmax(Q_j' K_i'^T / \sqrt{C})) V_i'), j \neq i \qquad (5)$$

**Coarse Matching.** Given the fusion features $\tilde{C_i}$, we adopt the LoFTR approach to generate coarse matches $M_c$. We apply a dual-softmax function to process a confidence matrix $P_c$, where $P_c(i, j)$ signifies the probability of pixel $i$ and pixel $j$ is a correct match. This probability takes into account both low-level image information

and high-level semantic information. By applying a threshold-based filter and a mutual nearest neighbor (MNN) search on $P_c$, we derive $M_c$.

## 3.3 Sub-pixel Refinement Module

As mentioned in Section 3.1, we also integrate the patch-wise refinement module from LoFTR for sub-pixel matching accuracy enhancement. Retrospect that the fine matching process utilized by LoFTR is oversimplified: For every coarse matching $(\tilde{i}, \tilde{j})$, LoFTR finds its position $(\hat{i}, \hat{j})$ on fine-level feature map $F_i$ and then crop two local windows based the location. After being transformed by the LoFTR module, for each local window only the center $\hat{i}$ of the enhanced feature is used to find its matched pixel $\hat{j}'$. When $\hat{i}$ is not within the overlap of two windows, this method may result in matching errors. As a supplement, we propose an overlap based fine-matching method. After the windows cropping, we calculate pixel-to-pixel similarities between the two windows. After applying threshold-based masking, a mutual nearest neighbor(MNN) search is conducted to refine the matches in the overlap area of windows, we represent the predictions for fine-level matches $M_f$ as:

$$M_f = \left\{ (\hat{i}', \hat{j}') \mid \hat{i}' = \mathcal{W}_{1 \to 0}(\hat{j}') \wedge \hat{j}' = \mathcal{W}_{0 \to 1}(\hat{i}') \right\} \quad (6)$$

where $\mathcal{W}$ is the warping operation, $\wedge$ is the and operation.

## 3.4 Self-supervised Training

Inspired by LoFTR, SRMatcher is trained using a self-supervised approach, eliminating the need for manual annotations. We create a geometrically transformed image $I_1$ from an original image $I_0$ by applying a predefined homography to $I_0$. Utilizing this homography, the true correspondence $j$ in $I_1$ for each element $i$ in $M_f$ can be precisely determined. Consequently, a set of ground-truth matches $G_f$ corresponding to $M_f$ is dynamically produced. Similarly, we generate ground-truth matches $G_c$ for $M_c$. Data augmentation includes random sampling of homographies. We also introduce various random photometric distortions to the image pairs, such as adjustments in brightness and contrast, motion blur, and Gaussian noise.

As shown in Figure 2, the coarse matching module, and the fine matching module each generate three pixel-to-pixel similarity matrices: $P_c$, and $P_f$, respectively. Ideally, these matrices should closely align with their respective ground truths. Following, the Focal binary cross-entropy Loss (FL) is employed to optimize multi-scale matching, with the loss defined as:

$$
\begin{aligned}
L_c &= -\frac{1}{|G_c|} \sum_{(i,j) \in G_c} log(P_c(i,j)) \\
L_f &= -\frac{1}{|G_f|} \sum_{(i,j) \in G_f} log(P_f(i,j))
\end{aligned}
\quad (7)
$$

The final loss is composed of the losses for the coarse-level and the fine-level, the cumulative loss function is expressed as: $L_{toatl} = L_c + L_f$.

## 4 EXPERIMENTS

### 4.1 Implementation Details

**Experimental data.** SRMatcher is trained on the outdoor Oxford-Paris dataset, adhering to the protocols established following [20].

This dataset is merged by the Oxford5K [25] and Paris6K [26] datasets, collectively referred to as Oxford-Paris, and encompasses a variety of outdoor and urban scenes. For evaluation, SRMatcher is tested on diverse image datasets, including HPatches [2], ISC-HE [20] and Megadepth [18]. These test datasets feature a range of image categories, from natural scenes with pronounced viewpoint and illumination shifts to extensively edited photographs. The training dataset's registered image pairs are generated automatically, ensuring the entire network training through a completely self-supervised approach.

**Implementation.** SRMatcher is developed in PyTorch. Given our computational resources (8 NVIDIA GeForce RTX 3090 GPUs), we set the larger image dimension to 640 for natural images during training. We use the Adam optimizer, with $\beta$ values of (0.9, 0.999) and a starting learning rate of 0.001. Each mini-batch consists of a pair of images. Training is capped at 15 epochs. SRMatcher is adapted for LoFTR [31] and its variations, including ASpan-Former [4] and GeoFormer [20], resulting in the creation of SR-Matcher_LoFTR, SRMatcher_ASpan, and SRMatcher_GeoFormer, respectively. The distinguishing feature among these three versions lies in the attention mechanism employed within the feature inter-action module, specifically linear attention [16], span attention [4] and geometrized attention [20]. For the inference phase, with the obtained fine matches $M_f$, we employ cv2.findHomography with RANSAC for reliable homography determination.

## 4.2 Evaluation on Natural Images

**Test Data.** The extensively utilized HPatches [2] dataset is employed, comprising 57 sequences with substantial illumination shifts and 59 sequences exhibiting significant viewpoint changes, making it a particularly demanding benchmark for homography estimation.

**Metrics.** Following [5, 31], we employ corner correctness to assess the accuracy of the estimated homography. The four corners from the first reference image are mapped to the second image using the estimated homography. The area under the cumulative curve (AUC) of the corner error is reported for threshold values of 1, 3, 5, and 10 pixels, respectively. For evaluation, all test images are resized to have their shorter dimension equal to 480.

**Comparative methods.** Matching approaches are categorized into two kinds of methods, i.e. detector-based feature matching and detector-free feature matching. We compiling a selection of 11 baseline methods as detailed below: 1) Detector-based feature matching methods including SIFT [22]+RooTSIFT [1], Superpoint [5], SuperGlue [30] and R2D2 [28]; 2) Detector-free feature matching methods including TopicFM [10], AdaMatcher [14], CasMTR [3], MESA [42], LoFTR [31], ASpanFormer [4], GeoFormer [20]. For LoFTR, ASpanFormer, GeoFormer, we test them with their original version and their upgraded version. For fair comparison we re-trained all these detector-free models using the same Oxford-Paris dataset. Due to the limitation of computing resources, we choose the 4c version of CasMTR with NMS.

**Result.** Table 1 displays the AUC scores for different methods, SRMatcher consistently exceeds the performance of other baseline methods across all error thresholds. Leveraging the formidable modeling prowess of Transformer, the attention-based matcher

**Table 1: Homography estimation results on HPatches. The best and second results are highlighted.**

| Method | Homography est. AUC | | | |
| --- | --- | --- | --- | --- |
| | @1px↑ | @3px↑ | @5px↑ | @10px↑ |
| *Detector-based matching* : | | | | |
| SIFT [22] | - | 46.3 | 57.4 | 70.3 |
| Superpoint [5] CVPRW'18 | - | 43.4 | 57.6 | 72.7 |
| R2D2 [28] NIPS'19 | - | 50.6 | 63.9 | 76.8 |
| SuperGlue [30] CVPR'20 | - | 53.9 | 68.3 | 81.7 |
| *Detector-free matching* : | | | | |
| TopicFM [10] AAAI'23 | 40.5 | 63.2 | 73.0 | 82.9 |
| AdaMatcher [14] CVPR'23 | 41.1 | 64.2 | 73.8 | 83.3 |
| CasMTR [3] ICCV'23 | 41.6 | 65.9 | 74.7 | 83.8 |
| MESA [42] CVPR'24 | 43.9 | 67.8 | 76.8 | 85.5 |
| LoFTR [31] CVPR'21 | 34.2 | 58.5 | 69.8 | 81.1 |
| SRMatcher_LoFTR | $38.6_{+12.87\%}$ | $62.3_{+6.49\%}$ | $72.1_{+3.29\%}$ | $82.3_{+1.48\%}$ |
| ASpan [4] ECCV'22 | 36.1 | 59.9 | 71.1 | 81.6 |
| SRMatcher_ASpan | $40.9_{+13.29\%}$ | $63.9_{+6.68\%}$ | $74.1_{+4.21\%}$ | $82.8_{+1.47\%}$ |
| GeoFormer [20] ICCV'23 | 44.3 | 68.0 | 76.8 | 85.4 |
| SRMatcher_GeoFormer | $\mathbf{49.2}_{+11.06\%}$ | $\mathbf{71.2}_{+4.71\%}$ | $\mathbf{79.3}_{+3.26\%}$ | $\mathbf{87.0}_{+1.87\%}$ |

demonstrates a notable lead over alternative methodologies. SRMatcher outperforms the previous SOTA GeoFormer in accuracy across various thresholds. SRMatcher demonstrates the most significant enhancement in AUC@1px, scoring 44.3 compared to 49.2. This superiority stems from the introduction of semantic constraints alongside geometric constraints. This integration helps eliminate unsound matches within the same set while strengthening matches based on semantic information. Qualitative results of homography estimation and matching are shown in Figure 4 and Figure 5. From the matching result of ISC-HE, we can see the all the matched point shared same semantic.

### 4.3 Evaluation on Manipulated Images

The models from Section 4.2 are utilized as is, without any further training.

**Test Data.** We follow [20] and use the ISC-HE datasets for the evaluation of manipulated images. Images were sourced from the Facebook AI Image Similarity Challenge (ISC) [7], where an original image undergoes various edits, like rotations and merges with other images, resulting in highly manipulated images. Given that ISC image pairs are unregistered, they were manually and collectively annotated, yielding 186 registered pairs. Each pair contains at least 8 correspondences.

**Results.** Table 2 illustrates that SRMatcher outperforms all baseline methods, even though the performance margin is smaller compared to the HPatches experiment. This is attributed to the highly challenging nature of ISC-HE. Unlike HPatches, ISC-HE displays forgery characteristics such as watermarks, cutouts, and image stitching, which pose challenges for semantic extraction in accurately capturing the image's semantics.

One thought-provoking finding from table 2 is the detector-based methods notably SuperGlue and SIFT outperform the advanced detector-baseline, i.e. ASpan. This result can be interpreted as follows: ISC-HE images underwent significant modifications, including watermark insertion and background replacement. Unlike

**Table 2: Homography estimation results on ISC-HE. The best and second results are highlighted.**

| Method | Homography est. AUC | | |
| --- | --- | --- | --- |
| | @3px↑ | @5px↑ | @10px↑ |
| *Detector-based matching* : | | | |
| Superpoint [5] CVPRW'18 | 18.3 | 39.0 | 62.2 |
| R2D2 [28] NIPS'19 | 18.2 | 39.6 | 62.9 |
| SIFT [22] | 19.9 | 42.4 | 65.0 |
| SuperGlue [30] CVPR'20 | 19.6 | 42.2 | 66.9 |
| *Detector-free matching* : | | | |
| TopicFM [10] AAAI'23 | 18.8 | 41.9 | 65.4 |
| AdaMatcher [14] CVPR'23 | 19.0 | 42.6 | 66.5 |
| CasMTR [3] ICCV'23 | 19.4 | 43.0 | 65.3 |
| MESA [42] CVPR'24 | 20.1 | 43.6 | 68.3 |
| LoFTR [31] CVPR'21 | 18.7 | 41.0 | 64.8 |
| SRMatcher_LoFTR | $19.2_{+2.67\%}$ | $41.4_{+0.97\%}$ | $65.1_{+0.46\%}$ |
| ASpan [4] ECCV'22 | 18.0 | 39.2 | 62.0 |
| SRMatcher_ASpan | $18.4_{+2.22\%}$ | $39.5_{+0.77\%}$ | $62.3_{+0.48\%}$ |
| GeoFormer [20] ICCV'23 | 19.9 | 43.8 | 68.4 |
| SRMatcher_GeoFormer | $\mathbf{20.5}_{+3.02\%}$ | $\mathbf{44.2}_{+1.01\%}$ | $\mathbf{68.8}_{+0.58\%}$ |

detector-free methods that rely on dense feature matching, detector-based approaches are less affected by these alterations. The superior performance of SRMatcher compared to both detector-based and detector-free methods represents the robustness and accuracy of semantic-aware feature matching methods.

### 4.4 Evaluation on Relative Pose Estimation

To demonstrate the applicability of our SRMatcher to other tasks, we test our model on the MegaDepth dataset for relative pose estimation. The models described in Section 4.2 are used directly, without undergoing additional training.

**Test Data.** MegaDepth includes 196 scene reconstructions from 1 million Internet images and 1500 pairs from two distinct scenes chosen for the test set, SRMatcher is tested in $1152 \times 1152$ following CasMTR [3].

**Metrics.** Following [42], we present the AUC of the pose error for thresholds ($5°$, $10°$, $20°$), where the pose error is determined by the maximal angular error of relative rotation and translation. In our evaluation approach, relative poses are deduced from the essential matrix, which is calculated from feature matching using RANSAC.

**Results.** As table 3 show, SRMatcher achieves the highest in AUC@$5°$ and AUC@$10°$ shows SRMatcher's strong task adaptability. It's important to note that MESA is designed specifically for relative pose estimation. Furthermore, SRMatcher offers significantly enhancement for GeoFormer, elevating the precision to the state-of-the-art level. This demonstrates that the semantic-aware matching provided by SRMatcher significantly augments the performance of feature matching.

**Table 3: Relative pose estimation results** (%) **on MegaDepth-1500 benchmark. The best and** second **results are highlighted.**

| Pose estimation AUC | MegaDepth1500 benchmark | | |
|---|---|---|---|
| | AUC@5° ↑ | AUC@10° ↑ | AUC@20° ↑ |
| TopicFM [10] AAAI'23 | 32.8 | 47.9 | 63.3 |
| AdaMatcher [14] CVPR'23 | 33.6 | 48.7 | 63.8 |
| CasMTR [3] ICCV'23 | 34.3 | 49.9 | 64.1 |
| MESA [42] CVPR'24 | **36.3** | 51.1 | 64.9 |
| LoFTR [31] CVPR'21 | 32.3 | 47.8 | 61.4 |
| SRMatcher_LoFTR | 33.8 | 49.1 | 63.0 |
| ASpan [4] ECCV'22 | 33.6 | 49.2 | 62.8 |
| SRMatcher_ASpan | 34.9 | 51.0 | 64.4 |
| GeoFormer [20] ICCV'23 | 34.2 | 50.5 | 63.5 |
| SRMatcher_GeoFormer | 35.8 | **52.0** | **65.2** |

## 4.5 Ablation Study

To assess the efficacy of our design, we perform an extensive ablation study on the components of SRMatcher on HPatches.

**Semantic-aware fusion block.** In order to demonstrated the significance of incorporating semantic information in the context of homography estimation, as row 1 of table 4 shows, without using semantic information (i.e., w/o SFB) we can see that the performance of AUC@1 decreases from 49.2 to 45.3. This outcome highlights the superior performance of the proposed semantic-aware feature representation learning framework. To show the impact of various fusion techniques utilized in SGIB, we compare our proposed SGIB against spatial attention (i.e., w/o SGIB). Comparing rows 2 and 7, it is evident that the feature produced by our SGIB outperforms the other, achieving the best performance. It indicates that our proposed SGIB by means of executes cross-attention between semantic and image features, fully leveraging the rich semantic information present in VFMs while preserving spatial details. More details are discussed in the supplementary.

**Semantic Extractors.** To further explore whether the fine-grained semantic features improve the effectiveness of matching results, we choose two separate semantic extractors. As Table 4 shows, we compare our approach with the ResNet-50, which is pretrained on ImageNet (i.e., w/ ResNet_50). The results are reported in row 3. Compared with row 7, it's evident that the semantic features produced by DINOv2 outperform the others in terms of effectiveness. This excellent performance is due to DINOv2's self-supervised training approach, which forces the model to learn image features that remain stable under different transformations, which tend to have a high semantic level. This result highlights the potential of integrating the more refined understanding of semantics, as features enriched with semantics provide robust representational ability.

**Overlap based fine matching.** To validate the usefulness of our overlap based fine matching, we use fine-matching2 in GeoFormer [20] as an alternative (i.e., w/o overlap based fine matching). As reported in row 6, we can see that the performance undergoes a significant drop. This result indicates the effects of our overlap based fine matching method.

**Table 4: Ablation study. Various variants of SRMatcher are evaluated for homography estimation on HPatches to highlight the significance of different components.**

| Modification | Homography est. AUC | | | |
|---|---|---|---|---|
| | @1px↑ | @3px↑ | @5px↑ | @10px↑ |
| w/o *SFB* | 45.3 | 69.3 | 77.8 | 86.0 |
| w/o *SGIB* | 47.8 | 70.1 | 78.5 | 86.0 |
| w/ *ResNet_50* | 47.2 | 69.7 | 78.3 | 86.2 |
| w/ *FPN* | 47.8 | 69.6 | 77.7 | 85.6 |
| w/o *Cross-images fusion* | 48.1 | 69.8 | 77.5 | 85.9 |
| w/o *Overlap based Fine Matching* | 47.6 | 70.3 | 78.6 | 86.3 |
| SRMatcher_GeoFormer | **49.2** | **71.2** | **79.3** | **87.0** |

**Fusion object.** In our matching network, the semantic features $S_0$ and $S_1$ are fused with the features $\hat{C}_0$ and $\hat{C}_1$, which are updated through self- and cross-attention. In this experiment, we try to verify the optimal fusion object. An alternative fusion type uses the coarse-level features $C_0$ and $C_1$. Before feeding into the self-attention and cross-attention mechanisms, we use the SFB to fuse the semantic features with the coarse-level features (i.e., w/ FPN). From row 4 of table 4, it is noted that fusion with coarse-level features $C_0$ and $C_1$ results in 47.8 of AUC@1, markedly lower than the 49.2 obtained when fusion with $\hat{C}_0$ and $\hat{C}_1$. These results indicate that $\hat{C}_0$ and $\hat{C}_1$ updated through the attention mechanism are better suited for fusion due to their enhanced position and context-related local features.

**Cross-images feature fusion.** To explore the effects of cross-images feature fusion strategy, we only utilize the operation of stage1 within SFB as mentioned in Section 3.2 (i.e., w/o cross-images fusion). This means that each image will only be fused with its own semantic features, regardless of semantic information from other images. The results are reported in row 5. Compared with row 7, it clearly show the usefulness of our cross-images feature fusion strategy. It's the same as human intuition, when humans perform matching operations, the area with the same semantics between two images should be paid attention to, because most matching points will be generated in this area. Therefore, it is necessary to pay attention to the area with the same semantics by referring to the semantics of other images.

## 5 LIMITATION AND DISCUSSION

Firstly, the inclusion of an additional semantic extractor and Semantic-aware Fusion Block (SFB) increases the computational complexity of SRMatcher compared to previous methods. Consequently, when processing large input images, the inference time of the model is prolonged. However, computational efficiency can be enhanced by optimizing the model's structure.

The generalization ability of the model is another critical aspect. When confronted with scientific data sets, such as medical images, DINOv2 may be unable to accurately identify image semantics. This is due to the limited training data set of DINOv2, which prevents generalization to more scenarios. In the future, constructing more extensive datasets for training could significantly improve its semantic extraction capabilities.

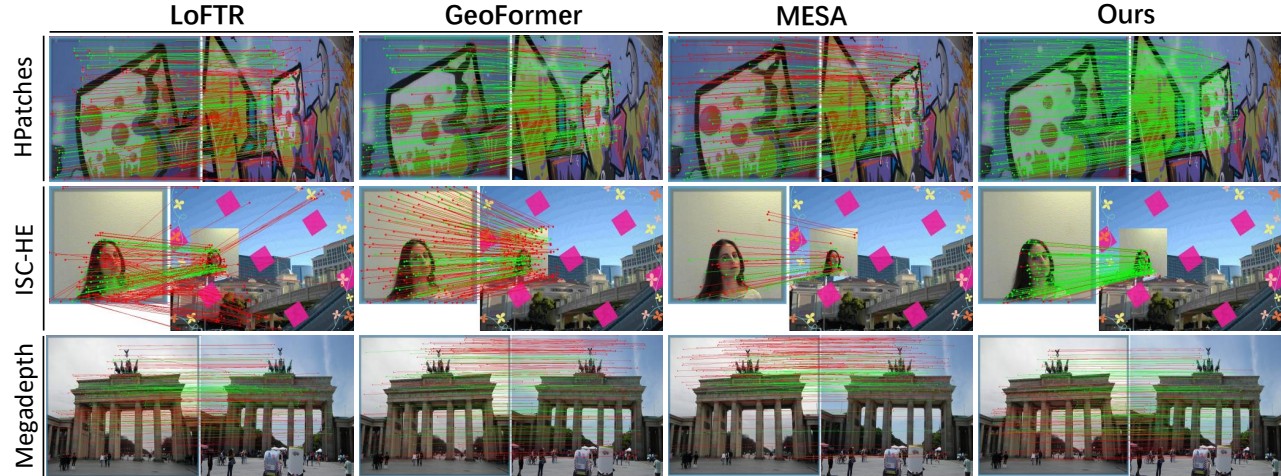

Figure 4: Qualitative of matching results with LoFTR [31], GeoFormer [20], MESA [42], and our SRMatcher. Points classified as inliers by RANSAC are displayed in green, while outliers are shown in red.

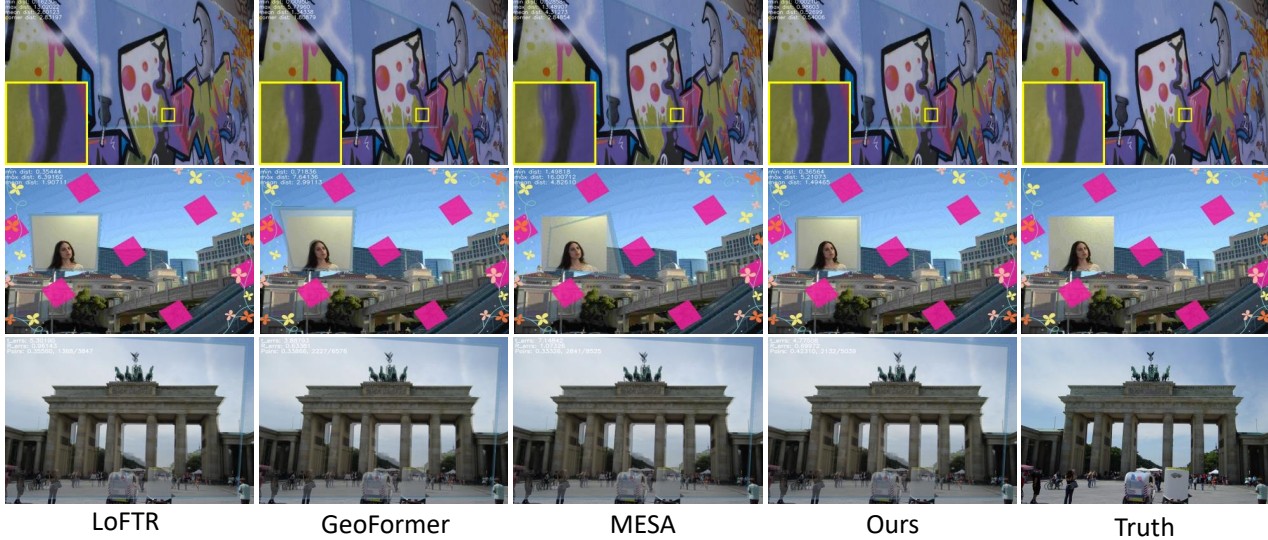

Figure 5: Qualitative of homography estimation results with LoFTR [31], GeoFormer [20], MESA [42], and our SRMatcher.

## 6 CONCLUSION

This work has proposed a novel detector-free feature matching method for homography estimation, named SRMatcher. We have discovered that previous matching networks only utilized coarse-grained semantic features, which prevents the full utilization of the knowledge that semantic extraction networks can offer and lacks adaptability with other tasks. Different from previous works, we explore a new way to integrate semantic information with the matching network. Specifically, we incorporate fine-grained image semantics derived from vision foundation models (VFMs), using our proposed Semantic-aware Fusion Block (SFB) to conduct cross-images feature interaction. This encourages our SRMatcher to to learn integrated semantic feature representation acquiring more accurate and realistic outcomes. Additionally, our SRMatcher can

be seamlessly integrated into the majority of LoFTR-based feature matching methods, where it consistently delivers outstanding results. Through comprehensive experimentation, we have explored the impact of our discoveries and highlighted the advantages of our proposed SRMatcher. We are confident that SRMatcher will provide new insights to the homography estimation community.

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
