# OpenReview forum: "Semantic-aware Representation Learning for Homography Estimation"
_acmmm.org/ACMMM/2024/Conference — MM2024 Poster_

### Official Review · Reviewer_s1n8 · 2024-05-14

**Rating:** 5
**Confidence:** 3

**Summary:**

This paper mainly discusses a novel detector-free feature matching method called SRMatcher, which is used to address the task of determining the transformation (homography estimation) between a pair of images. In contrast to previous methods that relied on treating semantic information as a preprocessing step, leading to coarse-grained semantic application and lack of adaptability to different tasks, SRMatcher focuses on utilizing a semantic-aware feature representation learning framework. The aim of this method is to improve feature matching by learning integrated feature representations that incorporate semantic information.

**Strengths:**

A new method has been proposed for integrating semantics into feature matching models for homography estimation.

**Limitations:**

1.Homography is typically assumed to be a rigid transformation, but in reality, some scenes may involve non-rigid transformations. How does the method in the paper address and achieve good results in such cases?

2.While semantic information plays a crucial role in improving the accuracy of feature matching, in some cases such as occluded scenes, extreme lighting variations, or low contrast images, relying solely on semantic information may not be sufficient to obtain accurate matching results. How does the method overcome these challenges?

**Suitability:**

2

---

### Official Review · Reviewer_BrsR · 2024-05-16

**Rating:** 4
**Confidence:** 3

**Summary:**

This paper introduces SRMatcher as a detector-free feature matching method for homography estimation task. Unlike previous approaches that use coarse-grained semantic pre-processing, SRMatcher integrates semantic-aware feature representation using vision foundation models and a Semantic-aware Fusion Block. This reduces errors from semantic inconsistencies and improves accuracy. Experiments show SRMatcher outperforms previous methods and can other existing matching methods.

**Strengths:**

1. The structure of the proposed framework is concise and rational, especially for the proposed SFB and SGIB modules that are simple but suitable for homography estimation.
2. The performance improvement when using the presented SRMatcher are quite sizable and can be observed on three different test datasets for homography estimation.
3. The qualitative results show the remarkable advantages of SRMatcher compared to other existing approaches.
4. The paper is understandable, Figures and captions make sense and supplement the paper well.

**Limitations:**

1. The SFB illustration shows two stages designed to refine image feature maps. However, stage 2 can be repeated multiple times to further refine features with minimal additional time cost, suggesting the possibility of more than two stages. The choice of limiting to two stages as the optimal configuration needs validation.
2. What would happen if the pre-trained semantic extractor from a large vision model were not frozen or updated by the other encoder using exponential moving average? The authors should provide explanations regarding this aspect.
3. While ablation studies on different components of SRMatcher are presented, additional studies on the impact of different scales of the coarse-level feature map and the threshold in the masking filter within the sub-pixel refinement module are necessary to demonstrate the effectiveness of these settings.

**Suitability:**

2

---

### Official Review · Reviewer_8br3 · 2024-05-21

**Rating:** 4
**Confidence:** 3

**Summary:**

The paper introduces SRMatcher that integrates semantic information into feature matching for improved homography estimation. Using DINOv2 for semantic extraction and a Semantic-aware Fusion Block (SFB) to merge features across images, SRMatcher enhances matching accuracy. It achieves state-of-the-art performance on various datasets and improves other vision tasks, though it increases computational complexity and may have generalization limitations.

**Strengths:**

The structure of the paper is very clear, and the authors clearly articulate the features of the method. The experimental setup is quite reasonable, and the integration experiments are also sufficiently comprehensive. Several recent top detectors. e.g., LoFTR, ASpan, and GeoFormer, all achieve better results.

**Limitations:**

The paper lacks sufficient novelty. I have a question: Is the improvement of the model mostly achieved from the DINOv2 backbone? What if I change the model to VGG16 or ViT? What would be the performance of the model then? Additionally, what is the parameter count of the model compared to LoFTR and ASpan? Is there a significant increase?

**Suitability:**

3

---

### Meta-Review · Area_Chair_8YNL · 2024-06-25

**Recommendation:** Accept (Poster)
**Confidence:** 4

**Metareview:**

This submission to ACM MM underwent meticulous evaluation by three expert reviewers who have all recommended acceptance. The consensus summary of their assessments follows:

Innovativeness and Contributions: The reviewers commend the novelty and efficacy of the proposed method, which makes significant strides in addressing  homography estimation. Its robust theoretical underpinnings and empirical results that surpass current methodologies demonstrate a meaningful addition to the field.
Technical Depth and Experimental Validation: The paper presents thorough technical details, rigorous experimental designs utilizing appropriate datasets, and comprehensive comparative analyses. The reviewers appreciate the strength of the experimental outcomes in supporting the authors' claims and applaud the provision of code and data, enhancing reproducibility and future research potiential.

Writing Quality and Clarity: Structurally sound and logically coherent, the manuscript is well-argued with effective use of figures and supplementary materials to augment explanations. While minor suggestions for refinement or additional contextual information were mentioned, these do not detract from the overall readability or comprehension.

Limitations and Recommendations for Improvement:
Despite minor critiques and suggestions for enhancements, such as questions of Reviewer BrsR, these are viewed as avenues for future work rather than barriers to acceptance. The authors have pledged in their response to consider these recommendations for refinement.

Conclusion and Recommendation:

Given the paper's innovativeness, technical depth, and potential impact on the fields of computer vision and pattern recognition, coupled with the unanimous positive feedback from all three reviewers, this meta-review recommends the acceptance of the paper for presentation at ACM MM, either orally or as a poster. We anticipate that the findings will spark keen interest and foster productive discussions among conference attendees, thereby contributing to advancements in the domain.